# Cutaneous Metastasis after Surgery, Injury, Lymphadenopathy, and Peritonitis: Possible Mechanisms

**DOI:** 10.3390/ijms20133286

**Published:** 2019-07-04

**Authors:** Isao Otsuka

**Affiliations:** Department of Obstetrics and Gynecology, Kameda Medical Center, Kamogawa 296-8602, Japan; otsuka.isao@kameda.jp

**Keywords:** cutaneous metastasis, skin metastasis, cancer, inflammation, wound healing, extranodal extension

## Abstract

Cutaneous metastases from internal malignancies are uncommon. Umbilical metastasis, also known as Sister Joseph nodule (SJN), develops in patients with carcinomatous peritonitis or superficial lymphadenopathy, while non-SJN skin metastases develop after surgery, injury, and lymphadenopathy. In this review, the possible mechanisms of skin metastases are discussed. SJNs develop by the contiguous or lymphatic spread of tumor cells. After surgery and injury, tumor cells spread by direct implantation or hematogenous metastasis, and after lymphadenopathy, they spread by extranodal extension. The inflammatory response occurring during wound healing is exploited by tumor cells and facilitates tumor growth. Macrophages are crucial drivers of tumor-promoting inflammation, which is a source of survival, growth and angiogenic factors. Angiogenesis is promoted by the vascular endothelial growth factor (VEGF), which also mediates tumor-associated immunodeficiency. In the subcutaneous tissues that surround metastatic lymph nodes, adipocytes promote tumor growth. In the elderly, age-associated immunosuppression may facilitate hematogenous metastasis. Anti-VEGF therapy affects recurrence patterns but at the same time, may increase the risk of skin metastases. Immune suppression associated with inflammation may play a key role in skin metastasis development. Thus, immune therapies, including immune checkpoint inhibitors reactivating cytotoxic T-cell function and inhibiting tumor-associated macrophage function, appear promising.

## 1. Introduction

The skin is a complex organ consisting of the epidermis, dermis, and skin appendages, including the hair follicle and sebaceous gland [1]. Cancer metastasis to the skin is uncommon and the incidence of skin metastases ranges from 1.0% to 4.6% in patients with internal malignancies [2,3]. The incidence of skin metastasis for different internal malignancies is variable: the most common primary tumors developing skin metastasis are breast and ovary in women and lung and colon in men [2].

Skin metastases usually develop at the umbilicus, surgical scars, including laparoscopic port sites, and in the vicinity of metastatic lymph nodes [4,5,6,7,8,9,10,11,12,13,14,15,16,17,18,19,20,21,22,23,24,25,26,27,28,29,30,31,32,33,34,35,36,37,38,39,40,41,42,43,44,45,46]. Skin metastases are often a late manifestation of the disease; however, in certain cases they may sometimes be the first sign of internal malignancies such as lung, renal, and ovarian cancers [47]. In this review, the possible mechanisms of skin metastases are discussed according to the site of appearance and the preceding medical conditions. Moreover, the roles of wound healing, inflammation, and adipose tissue in the development of skin metastases are also discussed. Skin involvement that develops as direct invasion from underlying primary tumors, such in breast or prostate cancer, is not discussed in this review.

## 2. Patterns of Skin Metastases

Based on the site of the lesion and the history of previous surgery, skin metastases are classified as metastatic umbilical tumors, which are known as Sister (Mary) Joseph nodules (SJNs), and non-SJN skin metastases. SJNs do not include umbilical metastases that develop as a port-site recurrence after laparoscopic surgery. Non-SJN skin metastases can further be divided into three major patterns, based on the preceding medical procedures or conditions as skin metastasis after surgery, injury, and lymphadenopathy (Table 1).

### 2.1. SJN

SJNs usually develop in patients with gastrointestinal and gynecological cancers [4,5,6,7,8,9,10]. SJN refers to a metastatic cancer of the umbilicus and is named after Sister Joseph, a nurse who frequently assisted Dr. William Mayo at St. Mary’s Hospital in Rochester, Minnesota, USA. She was the first person to observe that a firm umbilical nodule was often associated with intra-abdominal cancer: in a study evaluating 407 patients with SJNs, the most common origins are stomach (23%), ovary (17%), colon and rectum (15%), pancreas (9%), and uterus (6%) [48]. An SJN develops as either the first sign or a sign of recurrence in patients with peritoneal dissemination. Even though peritoneal dissemination responds completely to chemotherapy, an SJN could develop without accompanying recurrences [4]. In addition, SJNs could develop in patients with extensive involvement of the superficial lymph nodes, such as axillary and inguinal nodes [9].

### 2.2. Non-SJN Metastasis after Surgery

Skin metastases after surgery often develop at the site of surgical incision. Skin recurrences usually occur within the abdominal scar after surgery for gynecological and gastrointestinal cancers [4,5,11,12,13,14,15,16]. Similarly, many cases of port-site recurrences after laparoscopic surgery have been reported [5,21,22]. The estimated incidence of port-site recurrences in patients who underwent laparoscopic surgery for malignant disease is approximately 1–2% [49]. In rare cases, skin metastases at the site of surgical incision are the first sign of an undiagnosed cancer.

A skin recurrence can also occur at a surgical site remote from the primary tumor. For example, in patients with oropharyngeal cancer, skin metastases have occurred at percutaneous gastrostomy site [30], while skin metastases have developed in pacemaker pockets in patients with breast cancer [31].

Non-invasive tumors can also develop skin metastases. Borderline ovarian tumors, which lack destructive invasion microscopically, can metastasize to port sites after laparoscopic surgery [23]. Surgical scar endometriosis, i.e., an implant of normal endometrial tissue at the surgical incision, develops after cesarean section in 1–2% of patients [19] and it also develops at episiotomies and port sites [18,24].

### 2.3. Non-SJN Metastasis after Injury

Skin metastases could develop at the site of a traumatic injury, even though the site is remote from the primary tumor. In a patient with advanced prostate cancer who was treated with subcutaneous goserelin, skin metastasis developed at the injection site [35]. Skin metastasis at the site of the inflammatory response to skin test antigen (Dinitrochlorobenzene) developed in a patient with colon cancer [34]. In a patient with laryngeal cancer without lymph node metastasis, numerous superficial nodules developed circumscribed within the area that previously encased the body spica cast of a previous trauma [38].

### 2.4. Non-SJN Metastasis after Lymphadenopathy

After lymphadenopathy, skin metastases could develop in the vicinity of the metastatic superficial nodes. Skin metastases in the chest wall occurred after axillary node metastasis in patients with breast cancer. Skin metastases in the lower abdomen, scrotum, and penis occurred after inguinal node metastasis in patients with prostate cancer. Furthermore, skin metastases in the lower abdomen, vulva, and upper thigh occurred after inguinal node metastasis in patients with cervical cancer [39,40,43].

## 3. Possible Mechanisms of Skin Metastases

The process of skin metastasis involves two steps: the first step is the spreading of tumor cells to the skin. In SJNs, contiguous spread and lymphatic flow appear to be important. In non-SJN metastases, direct implantation, hematogenous metastasis, and extranodal extension play key roles (Figure 1). The second step is the proliferation of tumor cells at the site, which involves wound healing, inflammation, and the presence of adipose tissue.

### 3.1. Mode of Spread

#### 3.1.1. Several Routes to the Umbilicus

Tumor cells spreading to the umbilicus may occur via several routes. The most common mechanism is a contiguous spread, from the intraperitoneal metastasis to the umbilicus [7,50]. The vast majority of patients with metastases to the umbilicus, which is the thinnest part of the abdominal wall, have peritoneal dissemination. In addition, SJNs may be caused alternatively by spread through lymphatic channels or transvascular. However, these alternative mechanisms rarely cause the umbilical nodule, as there are no lymph nodes in and around the umbilicus and hematogenous dissemination in the absence of other sites of blood-borne metastasis is highly improbable [50]. However, it was reported that a patient without peritoneal dissemination with inguinal node metastasis developed an SJN [9]. In this case, an SJN may result from the alteration of the lymphatic flow by the tumor, which causes the obstruction of lymphatic pathways, shunting the lymphatic flow to the cutaneous lymphatics [42]. Obliterated umbilical arteries and the urachus could also provide pathways for tumor spread [7] and in a very rare case an SJN can occur via hematogenic pathway [10].

#### 3.1.2. Direct Implantation

The direct implantation of viable exfoliated tumor cells is the most likely mechanism for skin metastasis at surgical incisions and at laparoscopy port sites, where resected tumor tissues, both invasive and non-invasive, have passed during the procedure. Port site metastases can occur even at the trocar sites, where tumor tissues have not passed. Several mechanisms have been proposed for the development of port-site metastases. These mechanisms include direct wound contamination and implantation, the multiple effects of pneumoperitoneum, effects of the gases used for insufflation, “chimney effect,” aerosolization of tumor cells, and surgical technique [49]. Advanced malignancy and the presence of ascites may also be associated with port recurrences. Similarly, paracenteses in ovarian cancer with massive ascites, and needle biopsies for lung cancer can develop skin metastases by direct implantation [5,33].

Normal endometrial tissue, which is composed of endometrial glands with the surrounding stroma, can implant at the site of wounds formed by abdominal incision and episiotomy procedures. It seems possible that the direct implantation of benign tissues, including endometriosis, can occur when epithelial cells are supported by their stromal tissues supplying nutrients. As most successfully metastasizing tumors are those with the capability to induce stroma production in their new metastatic sites [51], endometriosis can metastasize even in the lung and/or in the pleura via the transdiaphragmatic passage [52].

#### 3.1.3. Hematogenous Metastasis

Cancer cells could spread to the skin via hematogenous routes. Skin metastases after surgery and injury usually develop in patients with advanced stage cancers, as circulating tumor cells (CTCs) are often detected in these patients. CTCs are found in frequencies on the order of 1–10 CTC per 1 mL of whole blood, which includes approximately 10 million leukocytes and 5 billion erythrocytes in cancer patients [53]. The detection of circulating tumor cells is a predictor of survival for patients with breast, colon, prostate, lung, and ovarian cancer [54,55,56,57,58]. However, patients with apparently early-stage cancer may develop hematogenous skin metastases after surgery, since hematogenous dissemination may occur very early in tumor development [59].

Skin metastases can develop not only at incisional scars after cancer surgery, but also at incisional scars after surgery for a benign disease performed in the presence of an undiagnosed cancer at the time of surgery. Skin metastases in patients with pancreatic and colon cancer have developed within surgical scars, where the surgery for a benign disease was performed one to twelve months before the cancer diagnosis [25,26,27,28,29]. These metastases may form by the colonization of cancer cells that are already present in the blood stream at the time of surgery for benign diseases, as these cancers tend to frequently develop hematogenous metastasis.

#### 3.1.4. Extranodal Extension

Cancer cells in the superficial lymph nodes can extend through the lymph node capsule into the surrounding subcutaneous adipose tissue. Patients with breast, prostate, and head and neck cancers tend to develop metastases to superficial lymph nodes (axillary, supraclavicular, and inguinal nodes) and often develop skin involvement [40,43]. In these cancers, extranodal extension is a prognostic factor for survival [60,61,62,63].

### 3.2. Mechanisms of Tumor Growth

#### 3.2.1. Wound Healing and Inflammation

After surgery and injury, the colonization and proliferation of tumor cells at the cutaneous tissues exploit the wound healing mechanisms. Many of the molecular mechanisms and signaling pathways are important for wound healing, and they have been involved in cancer cell proliferation [64]. Epithelial, endothelial, mesenchymal, and immune cells are interacted through growth factor/cytokine signaling pathways during wound healing and cancer progression [64,65].

Hemostasis is initiated immediately following injury: a fibrin clot is formed at the wound site to minimize blood loss. The inflammatory phase begins with blood coagulation and platelet activation. Growth and chemotactic factors such as platelet-derived growth factor (PDGF), insulin-like growth factor I (IGF-I), epidermal growth factor (EGF), and transforming growth factor-β (TGF-β) are released during this phase [65]. Lymphocytes, polymorphonuclear leukocytes (PMN), and monocytes/macrophage infiltrate the wound, in response to these factors [65]. For the migration of these cells, the fibrin plug is used as an interim matrix [64]. The following proliferative phase is characterized by angiogenesis and fibroblast proliferation [65]. In the remodeling phase, dermal fibroblasts/myofibroblasts actively reshape the dermal matrix by secreting collagen fibers and matrix metalloproteinases (MMPs) to restore it to pre-injury conditions [64]. In general, cancer-associated inflammation is characterized as a non-resolving condition; the inflammatory responses during wound healing being hijacked by tumor cells to facilitate tumor growth [66].

Inflammation is an immune response that is elicited by cellular damage due to noxious stimuli and conditions, such as infection and tissue injury. [67,68]. Inflammatory responses help an organism to restore tissue function and homeostasis through repair mechanisms [66]. Macrophages are crucial drivers of tumor-promoting inflammation by altering the tumor microenvironment and play critical roles in promoting metastatic invasion, proliferation, and survival of tumor cells through various mechanisms [69,70]. Macrophages are derived from circulating monocytes and recruited into tumors and are often the most abundant immune cells in the tumor. Based on their function and cytokine expression profile, macrophages are divided into two categories, or polarization: classical M1 and alternative M2 macrophages [66,71]. M1-polarized macrophages are pro-inflammatory and anti-tumorigenic, and activated by interferon gamma (IFN-γ). M2-polarized macrophages are anti-inflammatory, pro-tumorigenic, and activated by interleukin (IL)-4 [64,72]. Tumor-associated macrophages (TAMs) exhibit functions similar to those of the M2-polarized macrophages [71]. TAMs produce growth and survival factors for tumor cells (e.g., EGF, fibroblast growth factor [FGF], IL-6, and IL-8) and angiogenic factors (EGF, FGF, vascular endothelial growth factor [VEGF], PDGF, TGF-β, and CXC chemokines) and suppress the T-cell dependent antitumor immunity [73]. A distinct population of CD11b+ macrophages may recognize emigrating tumor cells and assist these cells with the invasion process [74]. Polarization of TAMs toward a M2 phenotype, as reflected by a lower M1/M2 ratio, is an independent predictor of shorter survival in locally advanced cervical cancer [75]. A recent study has reported that the presence of M1 macrophage in the tumor microenvironment increases the metastatic potential of ovarian cancer cells through the activation of the nuclear factor-κB signaling pathway by releasing tumor necrosis factor alpha (TNF-α) [76].

In addition, platelets can influence inflammation, immune regulation, and cancer progression, especially cancer metastasis. Platelets initiate hemostasis, inflammation, and wound healing. They are activated during chronic inflammation, cancer progression, and metastasis. Platelets invade into tumors with the use of leakiness or inflammatory reaction occurring during angiogenesis and help cancer cells to escape immune surveillance by adhering to them [77,78] Platelets also facilitate the arrest of disseminated tumor cells in the vascular system, and enhance invasive potential and extravasation of tumor cell [78].

Normal neutrophils mount a first defensive response against tumor cells and play major roles in linking inflammation and cancer, but they also actively involved in tumor progression and metastasis [79]. Neutrophils are the most abundant white blood cells in the circulation system and are the first responders in sites of acute tissue damage and infection. In a chronic inflammation context, neutrophils remain in tissues and this persistent presence is associated with cancer progression [80]. Neutrophils in the tumor microenvironment, also called tumor-associated neutrophils (TANs), are a heterogenous group of neutrophils consisting of antitumor phenotype (N1 TANs) and protumor phenotype (N2 TANs), analogous to M1/M2 macrophages [81]. N1 TANs directly target tumor cells and stimulate T cell immunity, and N2 TANs suppress T cell responses and upregulate angiogenic factors, such as VEGF [82]. Protumor functions, such as promoting tumor progression, invasion, metastasis, and angiogenesis appear to be preponderant [81].

Myeloid-derived suppressor cells (MDSCs) represent a heterogeneous population of immature myeloid cells and are able to suppress immune responses and stimulate tumor cell proliferation and angiogenesis [83]. MDSCs promote tumor growth by inhibiting the tumoricidal activity of T cells [84]. MDSCs also promote metastasis through promoting pre-metastatic niche formation [85]. Polymorphonuclear (PMN)-MDSCs and neutrophils share the same biological origin and many morphological and phenotypic features [86]. PMN-MDSCs directly interact with CTCs to promote their dissemination and enhance their metastatic potency [84].

Angiogenesis, which is the growth of new blood vessels from pre-existing vessels, is essential to cancer progression and wound healing, and is stimulated by inflammation. Angiogenesis plays a pivotal role in tumor growth by providing tumors with oxygen and nutrients and in tumor metastasis by providing a route for spread to distal tissues [87,88]. In acute wounds, VEGF, which is the most important pro-angiogenic factor, is secreted by neutrophils, macrophages, endothelial cells, and keratinocytes [64]. Following the initiation of angiogenesis, also called the “angiogenic switch,” tumors can more readily expand in size [63]. Low pericyte coverage and a hyperpermeable vasculature, driven by the overexpression of VEGF, can result in a more permissive environment for tumor cell intravasation, extravasation, and dissemination [89]. VEGF also inhibits the functional maturation of dendritic cells [90] and directly triggers regulatory T cell (Treg) proliferation [91], both of which are mechanisms that allow the escape of tumor cells from the host immune system. In addition, Tregs are actively recruited by tumors and suppress both adaptive and innate immune responses [92].

Fibroblasts and cancer cells are strongly interrelated in the tumor microenvironment [93]. Fibroblasts are activated in response to stimuli such as tissue injury. In cancer tissues, where persistent injurious stimuli exist, growth factors secreted by cancer cells stimulate the recruitment and activation of fibroblasts [94]. These cancer-associated fibroblasts (CAFs) produce IL-6, which promotes tumor growth by stimulating angiogenesis, cancer cell proliferation, and invasion. IL-6 also has inhibitory effects on immune cells [88,95]. CAFs are a major component of the cancer stroma, providing a fertile soil for tumor progression [93]. CAFs consist of different subpopulations with distinct functions and a subset of CAFs mediate chemoresistance [96].

#### 3.2.2. Adipose Tissue

The subcutaneous adipose tissue influences tumor growth, particularly in cases of direct implantation and extranodal extension [97]. In the tumor-surrounding adipose tissue, adipocytes at the tumor invasive front decrease in size and contain less lipids [98]. These cancer-associated adipocytes (CAAs) play a role as an energy source for cancer cells through the direct transfer of lipids to cancer cells [97]. CAAs also secrete several adipokines, such as TNF-α, IL-6, and IL-8, which support tumor cell growth [97]. The pre-existing inflammation such as radiation dermatitis and lymphangitis, often associated with lymphedema, may facilitate tumor growth in the adipose tissue. Chronic inflammation in the skin may alter angiogenesis and/or lymphangiogenesis [99], thus affect the development of skin metastasis.

### 3.3. Modifying Factors

Skin metastases are modified by host, tumor, and treatment factors.

#### 3.3.1. Host Factors

The most important host factor is age, which is the biggest risk factor for cancer. The aging process is linked to a gradual decline in the functional capacity of both adaptive and innate immune system, so called “immunosenescence” [100]. Immunosenescence appears to play a role in the development of distant metastasis. While SJN and non-SJN metastases that develop by direct implantation and extranodal extensions can occur both in the young and old, hematogenous skin metastases usually occur in the old (Table 1). In the old, anti-cancer immunity may be compromised because (i) the ability of neutrophils and macrophages to phagocytose pathogens decrease with aging, and (ii) the function of cytotoxic T cells is also compromised with aging [101]. Older individuals are also more susceptible to inflammatory diseases that promote tumor growth [102]. The numbers of MDSCs in the bone marrow and lymph nodes increase with aging in mice and MDSCs enhance the functions of other immunosuppressive cells, such as Tregs [100].

Obesity is a common cause of chronic inflammation and is strongly associated with poor prognosis in cancer patients [76,103,104]. A state of chronic inflammation in adipose tissue, which is observed in the majority of obese individuals, promotes cancer progression through accumulation of macrophages [76,104]. The metabolic syndrome, that includes hypertension, dyslipidemia, and insulin resistance, is associated with adipose inflammation and may promote tumor growth [104]. Thus obesity-associated chronic inflammation may be related to the development of tumor metastasis to the subcutaneous adipose tissue [76].

#### 3.3.2. Tumor Factors

The tumor factors associated with the development of skin metastases include the number of tumor cells at the metastatic site. For hematogenous metastasis, CTCs appear to play an important role—even though a hematogenous metastatic process is a highly inefficient process accomplished only by a minority of disseminated tumor cells [105], the probability of cancer cell survival increases as the number of cancer cells in the blood stream increases. For direct implantation, the continuous drainage of tumor cell-positive ascites through a port site is associated with the development of port-site metastasis.

The aggressiveness of the tumor cells is also an important factor. The rare metastatic cells arising as a result of selective pressure in the primary tumor have the ability to adopt migratory and invasive behavior [106]. Epithelial to mesenchymal transition (EMT) may also play a role in blood borne dissemination [105,107]: for example, cutaneous nasal metastasis has developed at presentation in a patient with undifferentiated ovarian carcinoma [46].

#### 3.3.3. Treatment Factors

The probability of developing skin metastases after cancer surgery depends on the surgical method. The probability is higher after laparoscopy than after open surgery [108]. Likewise, the probability is higher at smaller trocar sites than at larger trocar sites [109]. A previous study reported that the difference in recurrence rates between 5 mm and 10 mm diameter trocar sites was statistically significant [109]. An explanation for these observations is that tumor cell density, i.e., the number of tumor cells per unit volume, appears to be an important factor for direct implantation. In addition, the skin closure type affects the occurrence of implantation metastasis at the trocar incision scars. The incidence of recurrence at the trocar site have been statistically higher in patients undergoing a laparoscopy in which only the skin was closed at the end of the procedure than in the patients undergoing a laparoscopy with closure of all layers, i.e., the peritoneum, rectus sheath, and skin [110].

Adjuvant chemotherapy after surgery also affects the development of recurrences at surgical incision scars. In patients with gynecological cancer who underwent laparoscopic surgery, port-site recurrences developed only in patients who did not receive chemotherapy [109]. In patients with advanced ovarian cancer who underwent open laparoscopy, which is the separation of the different layers of the abdominal wall through a small incision (minilaparotomy), port-site metastases developed in 17% of the patients. However, all port-site metastases disappeared during primary therapy including chemotherapy [22]. Chemotherapy can eradicate tumor cells that colonize at surgical incisions, as the occurrence of surgical site metastasis of chemosenstive tumor, such as ovarian cancer, is uncommon.

Anti-VEGF antibodies, such as bevacizumab, may influence the development of skin recurrences. Although angiogenesis inhibitors that target the VEGF pathway may restrict tumor growth and metastatic ability [111], they concomitantly elicit tumor adaptation and progression to increased local invasion and distant metastasis occurrence [89]. Acquired resistance is common in VEGF-targeted therapies and the mechanisms that underlie the modest efficacy of anti-angiogenesis therapies may involve the active recruitment of macrophages to the tumor microenvironment, where they are responsible for the emergence of anti-VEGF therapy resistance [112].

## 4. Clinical Treatments

Inflammation that is associated with cancer may be a potential therapeutic target in patients with metastatic diseases [66,113]. Therapeutic targets that are involved in inflammatory responses include immune cells, such as TAM [69,70,71,114], neutrophils [79,80,81,82], MDSCs [83,84,115,116], as well as cancer-associated fibroblast and platelets [77,78,93,95,117,118]. Immune therapies involving immune checkpoint inhibitors, that reactivate cytotoxic T cell function and inhibit macrophage function, may also be promising [69,119,120,121]. Likewise, reducing Tregs, which suppress the anti-tumor immune response, with the concurrent activation of tumor-specific effector T cells will make the current cancer immunotherapy more effective [122].

Wound healing and inflammation enhance cancer stem cell (CSC) populations [123], therefore CSCs could also be a therapeutic target. CSCs have been recognized as the root of tumor drug resistance, recurrence, and metastasis [124]. Inhibition of CD47, that is highly expressed “don’t eat me” signal on cancer cells, particularly in CSCs [125], promotes the destruction of cancer cells by phagocytes such as macrophage and neutrophils [126]. As phagocytosis may result in antigen uptake and presentation, blockade of the CD47/signal regulatory protein alpha axis may synergize with immune checkpoint inhibitors that target the adaptive T-cell mediated immunity [127].

Immunotherapy is a promising therapeutic strategy for treating distant metastasis. Although effects of immunotherapy may be attenuated in old people due to immunosenescence, recent meta-analyses indicate that age may have little effect on efficacy of immune checkpoint inhibitors [128,129]. However, age-related changes in natural killer cells, which are involved in the efficient recognition of malignant target cells, may influence immune surveillance in a minority of elderly people [130]. As skin metastases often develop in obese individuals, adiponectin-based therapies inhibiting cancer advancement may provide a therapeutic approach to delay cancer progression in this type of patient [131].

## 5. Concluding Remarks

Metastases selectively develop in certain organs but not others, however tumor cells can reach the vasculature of all organs. This happens because malignant cells require a receptive microenvironment to engraft distant tissues and form metastases, which is stated in the “seed and soil” hypothesis [105,132]. Inflammation appears to alter tissues after traumatic injury into favorable microenvironments (‘niches’) for cancer cells, as the development of cancer metastases to inflammatory sites, such as surgical traumas, intestinal anastomosis, and bone fractures have been reported [133,134,135]. Comprehensive understanding of inflammation and cancer metastasis is needed to facilitate development of effective therapies to inhibit tumor progression in patients with metastatic cancer.

## Figures and Tables

**Figure 1 ijms-20-03286-f001:**
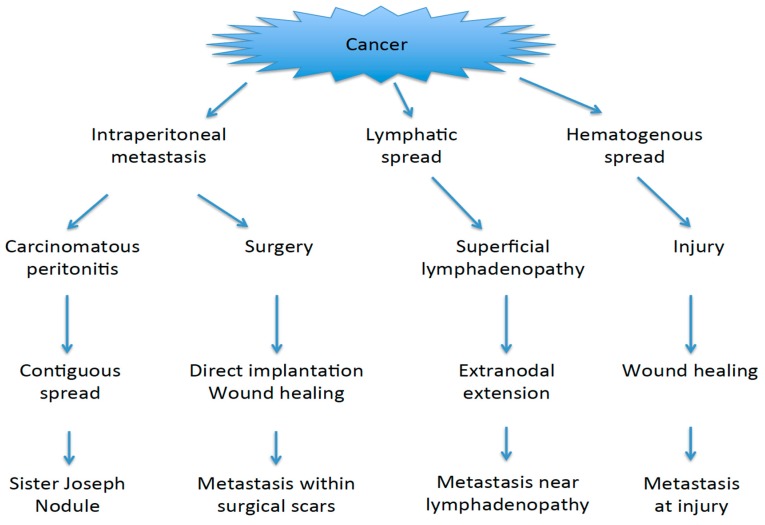
Possible mechanisms of skin metastases. Main pathways are depicted.

**Table 1 ijms-20-03286-t001:** Cases of cutaneous metastasis.

Preceding Condition or Procedure	Site of Cutaneous Metastasis	Mode of Tumor Spread	Primary Site of Tumors	Age	Ref.
***Umbilical Metastasis (Sister Joseph Nodule, SJN)***					
Carcinomatous peritonitis	Umbilicus	Direct invasion, Hematogenous, (Lymphatic)	Colon, Ovary, Stomach, Endometrium, Pancreas, Vulva, Appendix	32–95y	[4,5,6,7,8]
Inguinal node met	Umbilicus	Lymphatic, Hematogenous	Ovary, Colon	47, 83y	[9,10]
***Non-SJN Metastasis after Surgery***					
Surgical incision	Incision of tumor removal	Implant, (Hematogenous)	Breast, Ovary, Stomach, Colon, Cervix, Endometrium	25–68y	[4,5,11,12,13,14,15,16]
	Episiotomy for delivery	Implant	Cervix, Normal Endometrium	24–33y	[17,18]
	Incision of cesarean section	Implant	Normal Endometrium	21–40y	[19]
	Old scar before tumor removal	Implant, Hematogenous	Ovary	44y	[20]
Laparoscopy	Port site of tumor removal	Implant, Hematogenous	Ovary, Colon, Cervix, Endometrium, Gall Bladder	24–72y	[5,21,22]
		Implant	Borderline ovarian tumor, Endometriosis	28–37y	[23,24]
	Old scar before tumor removal	Hematogenous, Implant	Pancreas, Colon, Ovary	69–85y	[25,26,27,28,29]
PEG	Abdomen	Hematogenous	Head and Neck	50–72y	[30]
Pacemaker implantation	Breast	Hematogenous	Breast, Lung sarcoma	82, 89y	[31]
Drainage	Abdomen	Implant	Cervix, Ovary	37, 44y	[5,32]
Paracentesis	Abdomen	Implant, Hematogenous	Ovary	36–67y	[5]
***Non-SJN Metastasis after Injury***					
Needle biopsy	Chest wall	Implant, Hematogenous	Lung	79y	[33]
Injection	Arm, Abdomen	Hematogenous	Colon, Prostate	70, 74y	[34,35]
Puncture	Hand	Hematogenous	Cervix	54y	[36]
Abrasion	Vulva	Implant	Endometrium (after MIS)	87y	[37]
Body spica cast	Axilla	Hematogenous	Larynx	58y	[38]
***Non-SJN Metastasis after Lymphadenopathy***					
Inguinal node met	Lower abdomen, Inguinal area, Thigh	Lymphatic	Cervix, Prostate, Ovary	47–78y	[39,40,41,42]
Axillary node met	Chest wall	Lymphatic	Breast, Ovary	35–46y	[43,44]
Cervical node met	Chest wall, Face	Lymphatic	Cervix	41y	[45]
***Other Metastasis***					
Undifferentiated carcinoma	Nasal ala	Hematogenous	Ovary	58y	[46]

Cases of direct invasion of underlying carcinoma (breast, prostate cancer) to the skin are excluded. PEG, percutaneous endoscopic gastrostomy; met, metastasis; MIS, minimally invasive surgery.

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
