# Peer review of "Cutaneous Metastasis after Surgery, Injury, Lymphadenopathy, and Peritonitis: Possible Mechanisms"

_ijms, 2019, doi:10.3390/ijms20133286_

Round 1

Reviewer 1 Report

This is a well written and interesting manuscript – review regarding “Cutaneous Metastasis after Surgery, Injury, Lymphadenopathy, and Peritonitis: Possible Mechanisms”. However, there are some comments that could be considered as follow:

1. Another table summarizing the pathophysiological mechanisms of cutaneous metastases would be useful.

Author Response

Thank you for your comment. 

I added a new figure depicting “possible mechanisms of skin metastases.”

Reviewer 2 Report

good paper

Author Response

Thank you very much.

Reviewer 3 Report

Page 1, Line 16 - please use another word than hijacked

Table 1 - for SJN examples, there exists a case of SJN secondary to appendiceal carcinoma as well, which should be included in the manuscript. See PMID: 29719746

Line 115 - please fix the phrase "can be occur"

Line 182 - fix "that an organisms use" to singular or plural 

Author Response

In the abstract, I replace “hijack” with “exploit.” 

The paper in “Cureus 2018” are included in the reference.

Thank you for indicating grammatical errors. I have fixed this error.

This sentence was removed, because I rewrote this part.